# HypoSpace: Evaluating LLM Creativity as Set-Valued Hypothesis Generators under Underdetermination

## Abstract

As language models are increasingly used in scientific workflows, evaluating their ability to propose sets of explanations—not just a single correct answer—becomes critical. Many scientific problems are underdetermined: multiple, mechanistically distinct hypotheses are consistent with the same observations. We introduce HypoSpace, a diagnostic suite that treats LLMs as samplers of finite hypothesis sets and measures three complementary indicators: Validity (precision of proposals consistent with observations), Uniqueness (non-redundancy among proposals), and Recovery (coverage of the enumerated admissible set). We instantiate HypoSpace in three structured domains with deterministic validators and exactly enumerated hypothesis spaces: (i) causal graphs from perturbations, (ii) gravity-constrained 3D voxel reconstruction from top-down projections, and (iii) Boolean genetic interactions. Across instruction-tuned and reasoning-focused models, Validity often remains high while Uniqueness and Recovery degrade as the admissible space grows, revealing mode collapse that is invisible to correctness-only metrics. HypoSpace offers a controlled probe—rather than a leaderboard—for methods that explicitly explore and cover admissible explanation spaces. Code is available at: `https://anonymous.4open.science/r/HypoSpace-40DA`.

## 1 Introduction

Many scientific inference problems are *underdetermined* (Van Fraassen, 1980; Stanford, 2010): the same observations admit multiple, mechanistically distinct explanations. A capable scientific reasoning system should therefore not stop at finding one correct answer, but should map the complete admissible set $\mathcal{H}_O$ and surface multiple non-redundant hypotheses consistent with the data. As LLMs increasingly support scientific research, this capability becomes critical. Yet contemporary LLM evaluations largely reward one-shot correctness (Shojaee et al., 2025; Koblischke et al., 2025; Shojaee et al., 2024; Wang et al., 2024b; Coignion et al., 2024; Hendrycks et al., 2020), leaving open whether models can systematically explore sets of valid explanations under controlled underdetermination.

To address this gap, we introduce **HypoSpace**, a diagnostic suite that operationalizes classical creativity theory for LLM evaluation. Following established frameworks in divergent thinking research (Guilford, 1950; Torrance, 1966), which require creative outputs to be both *novel* and *appropriate* (Amabile, 1996), we define three complementary metrics for hypothesis generation: **Validity** enforces appropriateness by measuring consistency with observations; **Uniqueness** quantifies originality through non-redundancy among all proposals; and **Recovery** operationalizes fluency by measuring coverage of the enumerated admissible set $\mathcal{H}_O$. Unlike standard divergent-thinking tests, our framework provides *deterministic validators* and *enumerated ground truth*, eliminating rater subjectivity and enabling precise measurement.

Our approach treats LLMs as samplers that generate finite sets of candidate hypotheses (Figure 1a). For each problem instance, we enumerate the complete valid set $\mathcal{H}_O$, apply exact validity checks, and assess non-redundancy using task-specific canonicalizers that collapse semantically equivalent forms. By repeatedly sampling and measuring behavior along Validity (VR), Uniqueness (NR), and

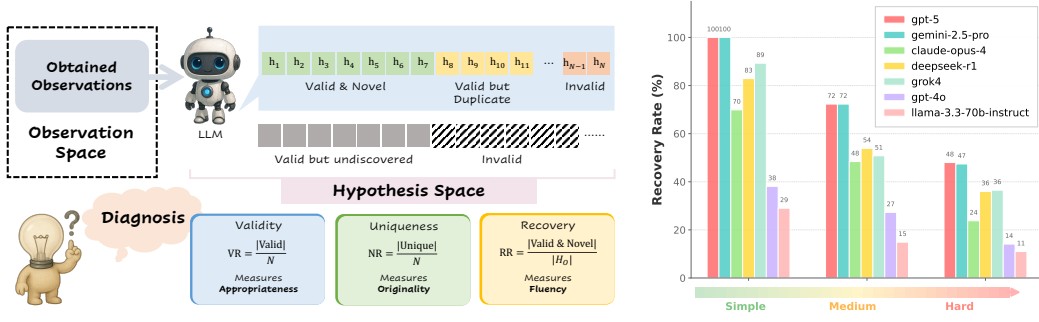

(a) HypoSpace Evaluation Framework.  (b) Model Comparison on RR.

Figure 1: **HypoSpace evaluation framework and model performance comparison.** (a) Our diagnostic approach treats LLMs as samplers over hypothesis spaces. Given observations $O$, models generate $N$ hypotheses that are validated for consistency with data and deduplicated for uniqueness. We measure three complementary indicators: Validity (VR: precision of valid hypotheses), Uniqueness (NR: non-redundancy among proposals), and Recovery (RR: coverage of the enumerated admissible set $\mathcal{H}_O$). These operationalize creativity theory's *appropriateness*, *originality*, and *fluency* dimensions, respectively. (b) Recovery Rate comparison across models on task of Boolean genetic interactions, showing systematic degradation as hypothesis spaces grow from simple to hard settings, with reasoning models generally outperforming non-reasoning models.

Recovery (RR), we decouple *being correct* from *exploring comprehensively*—a distinction obscured by traditional correctness-only metrics.

We instantiate this diagnostic suite across three structured domains that mirror scientific inference while enabling exact enumeration of valid hypothesis spaces: **Causal graphs from perturbations**, where models infer all DAGs consistent with single-node intervention observations; **3D voxel reconstruction under gravity**, where models reconstruct spatial configurations from top-down projections while satisfying physical constraints; and **Boolean genetic interactions**, where models propose expressions relating phenotype observations to underlying Boolean programs. Each domain provides natural difficulty scaling through controllable parameters (node counts, grid dimensions, operator complexity) that systematically vary the size of the admissible set $|\mathcal{H}_O|$. This controlled enumeration enables three key capabilities: direct measurement of hypothesis space coverage, precise calibration of task difficulty, and systematic analysis of sampling behavior including mode collapse patterns. Our goal is diagnostic measurement rather than leaderboard optimization—we seek to understand and improve how models explore solution spaces under underdetermination.

Evaluation across recent instruction-tuned and reasoning-focused LLMs reveals a consistent and concerning pattern: while models often maintain high **Validity** rates when generating admissible hypotheses, they exhibit pronounced *mode collapse* as hypothesis spaces grow. Both **Uniqueness** and **Recovery** degrade predictably with increasing $|\mathcal{H}_O|$ (Figure 1b), indicating that current models tend to circle a small subset of admissible explanations rather than systematically explore the complete space that observations allow. Crucially, because our valid sets are exactly enumerated, these coverage failures are measurable rather than anecdotal, and persist even when traditional accuracy metrics suggest strong performance. The primary contributions of this work are:

1. **Theoretical formulation:** We frame the evaluation of LLMs' ability to infer multiple distinct hypotheses fitting the same observations as *set-valued inference* under underdetermination, introducing three *diagnostic indicators* that systematically separate correctness from exploration capabilities. To the best of our knowledge, this is the very first exploration in this direction.

2. **Controlled diagnostic suite:** Three structured tasks with exact *enumeration* of valid hypothesis spaces, enabling non-LLM validity checking and objective measurement of coverage.

3. **Empirical findings:** A systematic study demonstrating that even frontier reasoning models exhibit *pronounced mode collapse*—high Validity coupled with degrading Uniqueness and

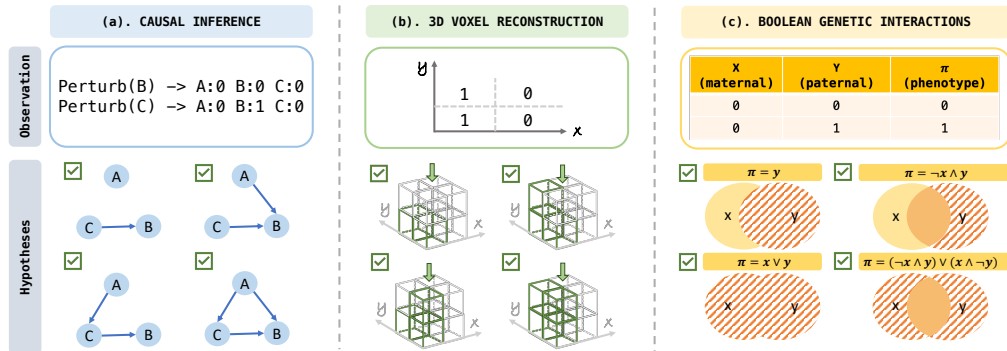

Figure 2: **HypoSpace task instantiations.** (a) *Causal inference from perturbations*: Given intervention observations (e.g., perturbing node C affects nodes B), models propose causal DAGs consistent with the data. Multiple valid graph structures can explain the same perturbation patterns. (b) *3D voxel reconstruction under gravity*: From top-down 2D projections, models reconstruct 3D voxel structures satisfying projection constraints and gravity (stacking rules). The same projection admits multiple valid 3D configurations. (c) *Boolean genetic interactions*: From phenotype observations of parental combinations, models propose Boolean expressions relating inputs to outputs. Expressions that collapse to the same form under our restricted canonicalizer count as one hypothesis. Each domain exemplifies scientific underdetermination where multiple mechanistically distinct hypotheses explain identical observations, enabling measurement of Validity, Uniqueness, and Recovery across enumerated hypothesis spaces.

Recovery as $|\mathcal{H}_O|$ increases. Our information-gain analysis versus query count further shows diminishing returns when the number of inference increases.

4. **Methodological contribution:** A reusable framework for analyzing hypothesis-generation capabilities, designed as a controlled probe for developing improved sampling strategies rather than a competitive benchmark.

We emphasize that HypoSpace does not claim to evaluate real-world scientific discovery. Rather, it abstracts core ingredients of set-valued inference—consistency checking and combinatorial hypothesis spaces—into controlled settings where ground truth is fully specified. This design choice prioritizes measurement precision and cross-model comparability.

## 2 RELATED WORK

Scientific-discovery benchmarks probe LLM reasoning in domains such as equation discovery and physics-inspired tasks, and in end-to-end AI-scientist loops (Shojaee et al., 2024; 2025; Koblischke et al., 2025; Coignion et al., 2024; Chen et al., 2025), but typically optimize for single-answer correctness and assume a single ground truth per input. Creativity-aware evaluations, which is rooted in novelty/appropriateness and the fluency/originality/flexibility triad (Guilford, 1950; Torrance, 1966; Amabile, 1996; Boden, 2004), have begun to incorporate diversity (e.g., HypoBench, IdeaBench, CreativEval, LiveIdeaBench) (Liu et al., 2025; Guo et al., 2025b; DeLorenzo et al., 2024; Ruan et al., 2024), and propose open-ended/axiomatic frameworks for originality and distributional creativity (Nagarajan et al., 2025; Wang et al., 2024a). Unlike these, we evaluate set-valued inference with deterministic validators and exactly enumerated admissible sets, enabling precise, model-agnostic measurement of Validity, Uniqueness, and Recovery without LLM-as-judge. (Extended related work in Appendix B.)

## 3 DIAGNOSTICS FOR CREATIVE HYPOTHESES GENERATION : DETAILS OF FORMULATION AND INDICATORS

**Problem setup.** Let $\mathcal{O}$ denote the full observation space and let $O \subseteq \mathcal{O}$ be the subset revealed for a given instance. Let $\mathcal{H}$ be the overall hypothesis space and $\mathcal{H}_O \subseteq \mathcal{H}$ the subset of hypotheses that

are consistent with $O$. Our diagnostic suite assumes that, for each instance, $\mathcal{H}_O$ is *explicitly enumerated*, enabling exact validity checks and direct measurement of coverage without LLM-as-judge circularity. We require three properties: (i) **soundness** (every $h \in \mathcal{H}_O$ is consistent with $O$), (ii) **completeness** (no valid hypothesis is omitted), and (iii) **Controllability** (the size of the ground-truth admissible set $|H_O|$ is controlled by task parameters—e.g., node count, grid dimensions, operator set/depth).

**Sampling protocol.** We evaluate models as *samplers over hypothesis sets*. From a fixed prompt and decoding setup, we draw $N$ independent samples $P = \{\tilde{h}_1, \ldots, \tilde{h}_N\}$, logging both the proposals and their order for novelty checks. In most experiments we set $N = |\mathcal{H}_O|$ to facilitate coverage analyses and RR@$k$ curves, but other sampling budgets are possible.

### 3.1 INDICATORS

We summarize behavior along three indicators that disentangle selection fidelity, non-redundancy, and coverage. Throughout, $\mathrm{val}_O(\tilde{h}) \in \{0, 1\}$ denotes the task-specific validity check and $\mathrm{nov}(\tilde{h}; A_{\tilde{h}}) \in \{0, 1\}$ indicates whether $\tilde{h}$ is distinct relative to the set of earlier samples $A_{\tilde{h}} \subseteq P$ (order-respecting).

**Validity (VR; appropriateness).**

$$\mathrm{VR}(P) \;=\; \frac{1}{N} \left| \{ \, \tilde{h} \in P \mid \mathrm{val}_O(\tilde{h}) = 1 \, \} \right|. \tag{1}$$

VR measures selection fidelity: the share of proposals that satisfy all observations.

**Novelty/Uniqueness (NR; originality).**

$$\mathrm{NR}(P) \;=\; \frac{1}{N} \left| \{ \, \tilde{h} \in P \mid \mathrm{nov}(\tilde{h}; A_{\tilde{h}}) = 1 \, \} \right|. \tag{2}$$

NR quantifies non-redundancy within a sampling run.

**Recovery (RR; fluency/coverage).**

$$\mathrm{RR}(P) \;=\; \frac{1}{|\mathcal{H}_O|} \left| \{ \, \tilde{h} \in P \mid \mathrm{val}_O(\tilde{h}) = 1 \wedge \mathrm{nov}(\tilde{h}; A_{\tilde{h}}) = 1 \, \} \right|. \tag{3}$$

RR measures coverage of the *enumerated* valid set and integrates validity and non-redundancy.

**Distinctness criteria.** Novelty/uniqueness is task-specific: labeled-edge equality (causal DAGs), voxelwise tensor equality (3D reconstructions), and a mechanistic canonicalizer for local Boolean equivalences (genetic interaction).

### 3.2 DIVERGENT-CREATIVITY VIEW (OPTIONAL ANALYSIS LENS)

Beyond our operational metrics (VR, NR, RR), we can view the hypothesis generation process from the perspective of *divergent creativity* in a Bayesian lens. Given an observation set $O$, each candidate hypothesis $h_i \in H$ has a posterior plausibility $P(h_i \mid O)$. Creativity in this sense reflects how *spread out* or *rich* the posterior distribution is across multiple admissible hypotheses. A simple measure is the number of distinct hypotheses supported above a plausibility threshold $\epsilon$:

$$C_{\mathrm{count}}(O) \;=\; \left| \{ h_i \in H : P(h_i \mid O) > \epsilon \} \right|,$$

which counts how many hypotheses remain comparably plausible given $O$. A complementary measure is the Shannon entropy of the posterior:

$$C_{\mathrm{entropy}}(O) \;=\; -\sum_{i=1}^{|H|} P(h_i \mid O) \log P(h_i \mid O),$$

which quantifies how evenly probability mass is distributed across hypotheses. High entropy indicates situations where many competing explanations are viable (greater divergent creativity), whereas low entropy reflects convergence toward a single dominant explanation. These measures conceptually align with our diagnostic indicators: RR as coverage/fluency, NR as originality, and VR as appropriateness.

### 3.3 INFORMATION-GAIN VIEW (SEQUENTIAL ANALYSIS LENS)

Although our diagnostics do not directly estimate entropies, they admit an information-theoretic view. Treat generation as a sequential process that grows a repertoire of mechanistic patterns. Let $\mathcal{H}_t$ be the multiset of hypotheses produced up to step $t$, $\mathcal{M}_t = M(\mathcal{H}_t)$ the induced set of patterns, and $p_t(m)$ the empirical frequency of pattern $m \in \mathcal{M}_t$. We define the stepwise information gain (entropy change) as

$$\Delta I_t \;=\; H_t - H_{t-1}, \qquad H_t := - \sum_{m \in \mathcal{M}_t} p_t(m) \log_2 p_t(m).$$

Positive $\Delta I_t$ indicates expanding pattern diversity (exploration), while negative values signal convergence toward repeated patterns. Empirically, sustained positive $\Delta I_t$ aligns with higher novelty and improved Uniqueness/Recovery.

### 3.4 INTENDED USE AND SCOPE

Our suite is a *diagnostic*, not a leaderboard: it isolates underdetermination, supports controlled ablations (sampling budget, model class), and reports interpretable indicators and coverage curves rather than a single scalar score. It does not claim real-world scientific discovery; instead, it provides a calibrated probe for set-valued inference.

## 4 HYPOSPACE CONSTRUCTION

We instantiate three structured diagnostics (Figure 2) that mirror common patterns of scientific inference while allowing exact enumeration of $\mathcal{H}_O$: causal graphs from perturbations, gravity-constrained 3D voxel worlds from top-down projections, and Boolean genetic interactions from phenotype observations.

### 4.1 CAUSAL INFERENCE FROM PERTURBATIONS

**Instance.** As in Figure 2a, let $V = \{v_1, \ldots, v_n\}$ be labeled nodes and $G^\star = (V, E^\star)$ a latent DAG. We observe single-node interventions and effects,

$$O = \{(s^{(k)}, x^{(k)})\}_{k=1}^m, \quad s^{(k)} \in V, \; x^{(k)} \in \{0,1\}^n,$$

with the forward model $F_G : V \to \{0,1\}^n$ given by

$$[F_G(v_i)]_j = \begin{cases} 0, & j = i \\ 1, & v_j \in \mathrm{desc}_G(v_i) \\ 0, & \text{otherwise.} \end{cases}$$

**Generation.** At each step, the model proposes a candidate DAG $\tilde{G} \leftarrow F_{\mathrm{LLM}}(O, A_{\tilde{G}}, T_{\mathrm{prompt}})$, where $A_{\tilde{G}}$ is the history.

**Validity and distinctness.** A proposal is valid iff it reproduces all observed effects: $\mathrm{val}_O(\tilde{G}) = 1 \Leftrightarrow F_{\tilde{G}}(s^{(k)}) = x^{(k)} \; \forall k$. Distinctness is on labeled nodes: two DAGs are identical if their labeled edge sets match; equivalently $\mathrm{Canon}(G) = \mathrm{Canon}(G')$.

**Scoring and difficulty.** Given $N$ samples $P = (\tilde{G}_i)_{i=1}^N$, we report VR/NR/RR as in Sec. 3. Difficulty is controlled by $n$ and $m$ (more nodes/interventions typically enlarge $|\mathcal{H}_O|$).

### 4.2 3D UNDERSTANDING UNDER GRAVITY

**Instance.** As in Figure 2b, an observation is a binary top-down projection $V \in \{0,1\}^{M \times M}$ where $V_{i,j} = 1$ indicates at least one occupied voxel in column $(i,j)$. Hypotheses are voxel stacks $\mathcal{H} = \mathcal{H}(M, K) \subseteq \{0,1\}^{K \times M \times M}$ with $K$ discrete layers (layer 1 bottom).

**Generation.** At each step, the model proposes $\tilde{h} \in \mathcal{H} \leftarrow F_{\mathrm{LLM}}(V, A_{\tilde{h}}, T_{\mathrm{prompt}}; M, K)$.

**Validity and distinctness.** A reconstruction is valid iff it satisfies projection and gravity (column-wise prefix) constraints:

$$\left(\forall i,j: \bigvee_{k=1}^{K} \tilde{h}_{i,j}^{(k)} = V_{i,j}\right) \wedge \left(\forall i,j,\ \forall k > 1: \tilde{h}_{i,j}^{(k)} \leq \tilde{h}_{i,j}^{(k-1)}\right).$$

Distinctness is voxelwise equality under aligned coordinates.

**Scoring and difficulty.** We report VR/NR/RR as in Sec. 3. Difficulty is controlled by grid size $M$, height budget $K$, and projection density; the number of valid completions grows rapidly with these parameters.

### 4.3 DNA INTERACTION VIA BOOLEAN PROGRAMS

**Instance.** As in Figure 2c, inputs are maternal/paternal phenotypes $x, y \in \{0, 1\}$ and output $\pi \in \{0, 1\}$. An instance specifies an operator set $\mathcal{F}$ (e.g., $\{\neg, \wedge, \vee\}$) and a depth bound $d$ defining the hypothesis space $\mathcal{H} = \mathcal{H}(\mathcal{F}, d)$ of expression trees over $\{x, y\}$ (optionally constants). Each $h \in \mathcal{H}$ induces $f_h : \{0, 1\}^2 \to \{0, 1\}$. Observations are pairs $O = \{((x^{(i)}, y^{(i)}), \pi^{(i)})\}_{i=1}^{m}$.

**Generation.** The model proposes $\tilde{h} \in \mathcal{H} \leftarrow F_{\text{LLM}}(O, A_{\tilde{h}}, T_{\text{prompt}}; \mathcal{F}, d)$.

**Validity and distinctness.** Validity requires functional agreement on observed pairs:

$$\text{val}_O(\tilde{h}) = 1 \iff f_{\tilde{h}}(x^{(i)}, y^{(i)}) = \pi^{(i)} \ \forall i.$$

Distinctness is defined by a mechanistic canonicalizer that collapses only local algebraic symmetries implemented in our code: commutativity, idempotence, and associativity flattening for repeated identical operators; two expressions are identical iff $\text{Canon}(h) = \text{Canon}(h')$.

**Scoring and difficulty.** We report VR/NR/RR as in Section 3. Difficulty is controlled by the operator set $\mathcal{F}$, depth $d$, and the number/coverage of observation pairs; these knobs govern $|\mathcal{H}_O|$ and the combinatorics of valid programs.

## 5 EXPERIMENTS

We evaluate a mix of instruction-tuned and "thinking" LLMs on three diagnostic tasks with enumerated valid sets: Causal Inference, 3D Understanding, and Boolean Genetic Interaction. The model suite includes GPT-4o (Hurst et al., 2024), GPT-5 (OpenAI, 2025), Gemini-2.5-Pro (Comanici et al., 2025), Claude-Opus-4 (Anthropic, 2025), DeepSeek-R1 (Guo et al., 2025a), LLaMA-3.3-70B-Instruct (Touvron et al., 2023), and Grok-4 (xAI, 2025). [1] We group models into **reasoning** (a.k.a. "thinking") and **non-reasoning** (instruction-tuned) categories. Reasoning models, by default, produce explicit intermediate rationales and are marketed as reasoning-optimized; non-reasoning models typically return short direct answers unless prompted otherwise. We score only the final structured hypothesis, not the rationale text. We report Validity (VR), Novelty/Uniqueness (NR), and Recovery (RR) as defined in Section 3.

### 5.1 EXPERIMENTAL DETAILS AND DIFFICULTY

Each task exposes natural knobs that scale the size of the admissible set $|\mathcal{H}_O|$. **Causal Inference** varies the number of nodes and observed interventions; **3D Voxel Reconstruction** varies the number of top-down views at fixed height budget; **Boolean Genetic Interaction** varies the Boolean operator set/depth and observation coverage. We use three regimes (simple/medium/hard). For transparency, the instance settings are printed in the Table 1. Unless noted otherwise, for each instance we draw $N$ independent samples with $N = |\mathcal{H}_O|$, compute VR/NR/RR on the $N$ proposals, and aggregate by averaging across instances within task/difficulty (reporting mean±std across repeated runs).

**Validation and distinctness.** Outputs follow strict schemas and are checked by deterministic validators (forward simulation for causal DAGs; projection and gravity for 3D; functional agreement on

---

[1] All models were accessed via OpenRouter OpenRouter (2025), except `GPT-5`, which was evaluated via the OpenAI API. To preserve double-blind review, provider/version metadata and full prompts/decoding settings are deferred to the appendix C. Our total API budget was $\sim \$1,000$.

Table 1: **HypoSpace evaluation.** Performance across reasoning and non-reasoning models using three complementary metrics: Validity Rate (VR; precision of hypotheses consistent with observations), Uniqueness/Novelty Rate (NR; non-redundancy among proposals), and Recovery Rate (RR; coverage of the enumerated admissible set $\mathcal{H}_O$). Settings scale from Difficulty level 1 (simple) to 3 (hard) with corresponding increases in hypothesis space size $|H_O|$. Results show mean $\pm$ standard deviation across instances. Reasoning models maintain high performance across all metrics, while non-reasoning models show degraded VR and consequently lower RR, particularly in harder settings with larger hypothesis spaces. We highlight the best and second-best results in the table.

| Difficulty | Metric | Reasoning Models | | | | | Non-Reasoning Models | |
|---|---|---|---|---|---|---|---|---|
| | | gpt-5 | gemini-2.5-pro | claude-opus-4 | deepseek-r1 | grok4 | gpt-4o | llama-3.3-70b-instruct |
| **Task 1: Causal Inference** | | | | | | | | |
| 1 (nodes=4) | VR | 100.00% ± 0.00% | 100.00% ± 0.00% | 89.30% ± 21.70% | 100.00% ± 0.00% | 100.00% ± 0.00% | 73.80% ± 32.70% | 30.00% ± 40.10% |
| | NR | 100.00% ± 0.00% | 100.00% ± 0.00% | 86.20% ± 20.80% | 98.20% ± 4.50% | 100.00% ± 0.00% | 83.20% ± 23.00% | 83.30% ± 27.00% |
| | RR | 100.00% ± 0.00% | 100.00% ± 0.00% | 77.50% ± 27.10% | 98.20% ± 4.50% | 100.00% ± 0.00% | 60.90% ± 34.50% | 24.00% ± 34.40% |
| 2 (nodes=5) | VR | 100.00% ± 0.00% | 100.00% ± 0.00% | 80.10% ± 23.10% | 99.00% ± 3.30% | 100.00% ± 0.00% | 49.00% ± 36.20% | 17.60% ± 28.10% |
| | NR | 100.00% ± 0.00% | 100.00% ± 0.00% | 79.90% ± 23.60% | 93.50% ± 12.00% | 100.00% ± 0.00% | 83.50% ± 24.60% | 86.00% ± 21.00% |
| | RR | 100.00% ± 0.00% | 100.00% ± 0.00% | 65.40% ± 28.40% | 92.50% ± 12.80% | 100.00% ± 0.00% | 38.30% ± 32.80% | 17.60% ± 28.10% |
| 3 (nodes=6) | VR | 100.00% ± 0.00% | 99.80% ± 0.80% | 59.30% ± 21.20% | 98.20% ± 3.00% | 100.00% ± 0.00% | 72.80% ± 34.10% | 10.40% ± 26.00% |
| | NR | 99.20% ± 1.40% | 99.40 % ± 1.30% | 90.00% ± 7.90% | 81.20% ± 9.80% | 99.80% ± 1.20% | 22.90% ± 27.20% | 38.00% ± 22.40% |
| | RR | 99.20% ± 1.40% | 99.20% ± 1.40% | 51.10% ± 22.10% | 79.50% ± 9.30% | 99.80% ± 1.20% | 6.70% ± 8.30% | 2.30% ± 2.70% |
| **Task 2: 3D Understanding** | | | | | | | | |
| 1 (tp=1) | VR | 100.00% ± 0.00% | 100.00% ± 0.00% | 100.00% ± 0.00% | 96.30% ± 10.50% | 100.00% ± 0.00% | 63.00% ± 24.60% | 18.50% ± 16.60% |
| | NR | 100.00% ± 0.00% | 100.00% ± 0.00% | 100.00% ± 0.00% | 100.00% ± 0.00% | 100.00% ± 0.00% | 96.30% ± 10.50% | 88.90% ± 15.70% |
| | RR | 100.00% ± 0.00% | 100.00% ± 0.00% | 100.00% ± 0.00% | 96.30% ± 10.50% | 100.00% ± 0.00% | 59.30% ± 26.20% | 18.50% ± 16.60% |
| 2 (tp=2) | VR | 100.00% ± 0.00% | 100.00% ± 0.00% | 90.40% ± 12.40% | 99.30% ± 2.80% | 100.00% ± 0.00% | 38.50% ± 19.20% | 10.40% ± 16.70% |
| | NR | 100.00% ± 0.00% | 100.00% ± 0.00% | 87.80% ± 12.30% | 99.60% ± 2.00% | 100.00% ± 0.00% | 75.20% ± 14.80% | 84.40% ± 13.30% |
| | RR | 100.00% ± 0.00% | 100.00% ± 0.00% | 78.10% ± 11.30% | 98.90% ± 3.30% | 100.00% ± 0.00% | 30.70% ± 12.70% | 8.90% ± 13.60% |
| 3 (tp=3) | VR | 100.00% ± 0.00% | 99.90% ± 0.70% | 62.30% ± 19.70% | 99.00% ± 1.60% | 99.90% ± 0.70% | 19.40% ± 9.30% | 10.50% ± 13.10% |
| | NR | 98.80% ± 2.20% | 95.20% ± 4.60% | 81.10% ± 10.60% | 86.20% ± 5.20% | 99.90% ± 0.70% | 57.90% ± 14.00% | 66.70% ± 14.30% |
| | RR | 98.80% ± 2.20% | 95.10% ± 4.80% | 48.70% ± 11.50% | 85.20% ± 4.90% | 99.80% ± 0.90% | 14.30% ± 6.50% | 6.00% ± 5.60% |
| **Task 3: DNA Interaction** | | | | | | | | |
| 1 (basic) | VR | 100.00% ± 0.00% | 100.00% ± 0.00% | 88.30% ± 16.50% | 83.90% ± 15.50% | 89.30% ± 21.70% | 88.40% ± 26.20% | 95.60% ± 18.70% |
| | NR | 100.00% ± 0.00% | 100.00% ± 0.00% | 69.90% ± 19.40% | 82.90% ± 16.30% | 89.30% ± 21.70% | 38.10% ± 21.40% | 30.10% ± 14.70% |
| | RR | 100.00% ± 0.00% | 100.00% ± 0.00% | 69.90% ± 19.40% | 82.90% ± 16.30% | 89.30% ± 21.70% | 38.10% ± 21.40% | 29.00% ± 15.70% |
| 2 (extended) | VR | 74.90% ± 21.40% | 72.30% ± 21.10% | 52.70% ± 23.20% | 57.20% ± 21.70% | 52.20% ± 20.40% | 83.00% ± 27.50% | 55.80% ± 45.50% |
| | NR | 74.90% ± 21.40% | 72.30% ± 21.10% | 54.90% ± 24.50% | 57.20% ± 21.70% | 52.20% ± 20.40% | 31.60% ± 21.50% | 18.40% ± 16.70% |
| | RR | 72.30% ± 21.30% | 72.30% ± 21.10% | 48.40% ± 23.20% | 53.90% ± 24.50% | 50.80% ± 20.70% | 27.30% ± 22.90% | 14.90% ± 17.80% |
| 3 (full) | VR | 65.10% ± 12.50% | 52.20% ± 15.50% | 36.90% ± 15.90% | 41.00% ± 13.60% | 40.60% ± 15.00% | 68.10% ± 37.40% | 66.60% ± 43.90% |
| | NR | 49.90% ± 14.00% | 48.90% ± 14.90% | 24.20% ± 10.20% | 36.60% ± 12.30% | 37.90% ± 15.20% | 21.10% ± 14.30% | 14.30% ± 9.90% |
| | RR | 48.00% ± 14.40% | 47.40% ± 13.40% | 23.80% ± 10.20% | 36.00% ± 12.80% | 36.50% ± 14.90% | 14.10% ± 10.50% | 11.00% ± 9.40% |

observed pairs for DNA). Distinctness is assessed via labeled-edge equality (causal), tensor equality (3D), and a mechanistic canonicalizer that collapses local algebraic symmetries (DNA).

## 5.2 RESULTS BY TASK

**Results.** Table 1 summarizes Validity (VR), Uniqueness (NR), and Recovery (RR) across difficulty for the three diagnostics. In causal inference, simple/medium regimes are near-ceiling for several reasoning models; in the hard case (6 nodes) the top reasoning models sustain near $100\%$ VR and high NR/RR, while other reasoning models show small but consistent NR/RR gaps and non-reasoning models lag in VR (hence RR). In 3D voxel reconstruction, most models attain high VR with 1–2 views, but at 3 views ($|\mathcal{H}_O|{=}27$) gaps widen: frontier reasoning models preserve NR/RR whereas others repeat proposals early, depressing RR. Boolean genetic interactions is most discriminative: as operator set/depth and observation coverage grow, NR and RR drop markedly for all models even when VR remains moderate; the canonicalizer collapses superficial variants, revealing limited exploration of distinct mechanisms.

## 5.3 CROSS-TASK OBSERVATIONS

Across all three domains we see the consistent trend: as $|\mathcal{H}_O|$ grows, Uniqueness and Recovery drop even while frontier reasoning models (GPT-5, Gemini-2.5-Pro, Claude-Opus-4, DeepSeek-R1, Grok-4) remain high on Validity. By task, Causal Inference is most tractable at our scales (several

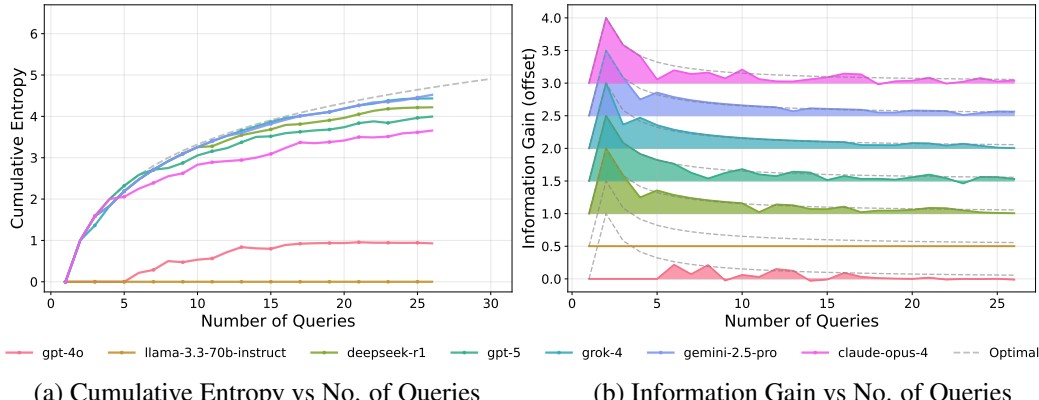

(a) Cumulative Entropy vs No. of Queries      (b) Information Gain vs No. of Queries

Figure 3: **Information-theoretic analysis of hypothesis space exploration in HypoSpace.** **(a)** Cumulative entropy as a function of sequential queries, measuring how uncertainty reduction varies across models as they sample from the hypothesis space. Steeper curves indicate more efficient exploration of distinct valid hypotheses. **(b)** Information gain per query (with 0.5 vertical offset for visual clarity). The curve shape reflects each model's marginal contribution of new information with additional samples, revealing differences in exploration strategies and susceptibility to mode collapse. Models with flatter curves show diminishing returns in hypothesis diversity, consistent with the Recovery Rate degradation observed in Table 1.

models near ceiling), 3D Voxel is intermediate with collapse emerging in the hardest multi-view settings, and Boolean Genetic Interactions is most discriminative—its large program space, even after canonicalization, yields the sharpest coverage deficits. Reasoning-capable models consistently beat non-reasoning baselines on NR/RR at medium–hard difficulties, indicating that explicit reasoning mitigates—but does not eliminate—mode collapse.

## 5.4 WHERE COVERAGE FAILS: EVIDENCE FOR MODE COLLAPSE

The systematic decline in RR across increasing task difficulty provides direct evidence for mode collapse in hypothesis generation, where models converge on limited subsets of the admissible space despite maintaining solution validity. This collapse manifests through three complementary mechanisms: models exhibit strong attraction to a small number of preferred hypotheses, leading to early saturation where additional sampling yields diminishing returns in unique valid discoveries; they demonstrate systematic blindness to structural symmetries, particularly evident in Boolean Genetic Interaction where syntactically diverse expressions collapse under canonicalization; and they show sensitivity to sampling budget constraints, with coverage growing sub-linearly even when the number of samples approaches the size of the valid set.

## 5.5 INFORMATION-THEORETIC ANALYSIS OF EXPLORATION DYNAMICS

Figure 3 provides an information-theoretic lens on hypothesis space exploration, revealing fundamental differences in how models navigate the enumerated admissible sets. The cumulative entropy curves (Figure 3a) demonstrate that frontier reasoning models achieve steeper entropy growth, indicating more efficient discovery of distinct valid hypotheses as sampling progresses. In contrast, non-reasoning models plateau earlier, reflecting convergence to smaller hypothesis subsets. The information gain analysis (Figure 3b) further illuminates these exploration dynamics: reasoning models maintain higher marginal information contributions per query, while non-reasoning models exhibit flatter curves characteristic of diminishing returns. This pattern directly correlates with the Recovery Rate degradation observed in Table 1, providing mechanistic evidence that mode collapse manifests as premature entropy saturation rather than uniform exploration inefficiency.

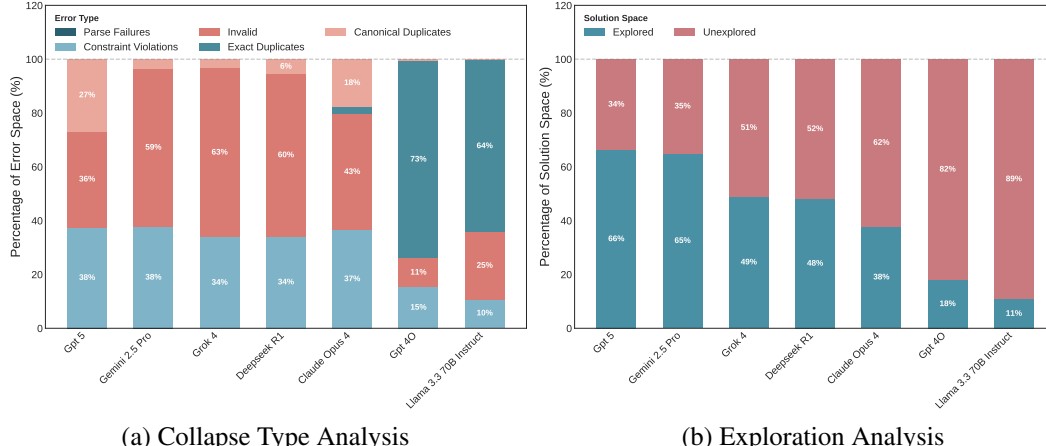

(a) Collapse Type Analysis
(b) Exploration Analysis

Figure 4: **Failure modes and exploration patterns in HypoSpace hypothesis generation. (a)** Error categorization across models: Distribution of failure types including parse failures (malformed outputs), constraint violations (structurally invalid hypotheses), invalid generations (inconsistent with observations), and duplicates (exact and canonical equivalences). Higher proportions of duplicates indicate mode collapse tendencies. **(b)** Hypothesis space coverage: Fraction of the enumerated admissible set $\mathcal{H}_O$ explored versus unexplored by each model. Limited exploration (lower blue bars) corresponds to reduced Recovery Rates and reveals the extent to which models fail to map the full space of valid explanations, even when maintaining high Validity.

## 5.6 FAILURE MODE DECOMPOSITION AND COVERAGE ANALYSIS

Figure 4 decomposes the sources of limited hypothesis coverage through complementary analyses of failure types and exploration completeness. The error categorization (Figure 4a) reveals that mode collapse stems primarily from duplicate generation rather than validity failures: while parse errors and constraint violations remain relatively low across models, exact and canonical duplicates constitute the dominant failure mode, particularly for non-reasoning models. This pattern indicates that models can generate structurally valid hypotheses but struggle to diversify beyond preferred templates. The exploration analysis (Figure 4b) quantifies this limitation directly, showing that even high-performing models like GPT-5 and Gemini-2.5-Pro explore only 60-70% of enumerated admissible sets, with weaker models covering substantially smaller fractions. The systematic under-exploration persists despite models maintaining high Validity rates, confirming that current LLMs exhibit fundamental constraints in mapping complete hypothesis spaces under underdetermination.

## 6 DISCUSSION

As Table 1 shows, indicators worsen as $|\mathcal{H}_O|$ grows: Causal saturates quickly for top models, while 3D—and especially Boolean—remain discriminative at higher difficulty. Reasoning models consistently beat non-reasoning baselines on NR/RR even when VR is similar on easy cases. The three tasks probe complementary skills (structural, spatial, symbolic) and expose distinct collapse modes (edge-pattern fixation; minimal-height towers; algebraic template reuse).

## 7 CONCLUSION

We introduced a *diagnostic suite* for *set-valued* evaluation of LLMs under underdetermination, with exact validators and enumerated valid sets. Across three structured tasks, frontier models show **high Validity** but limited **Uniqueness/Recovery** as the admissible set grows. Because VR/NR/RR are measured against enumerated ground truth (Table 1), the observed collapse is quantitative, not anecdotal. We view these diagnostics as a controlled probe set—not a leaderboard—for developing methods that *map* admissible hypothesis spaces (e.g., entropy-seeking decoding, explicit memory across samples, or learned proposal distributions).

## REPRODUCIBILITY STATEMENT

We will release the full codebase, including task generators, exact admissible-set enumerators, deterministic validators/canonicalizers, prompts and decoding configs.

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

# APPENDIX

## A  ANALYSIS OF COST AND TOKENS ACROSS LLMS

Table 2: Model Cost Efficiency Analysis

| Model | Total Cost ($) | Total Tokens | Cost/1M Tokens ($) | Avg Tokens/Run |
|---|---|---|---|---|
| claude-opus-4 | 3.51 | 3,511,372 | 1.00 | 390,152 |
| deepseek-r1 | 41.45 | 22,701,210 | 1.83 | 2,522,356 |
| gemini-2.5-pro | 145.88 | 16,789,228 | 8.69 | 1,865,469 |
| gpt-4o | 10.66 | 2,886,161 | 3.69 | 320,684 |
| gpt-5 | 93.25 | 11,510,695 | 8.10 | 1,278,966 |
| grok-4 | 22.21 | 22,214,056 | 1.00 | 2,468,228 |
| llama-3.3-70b-instruct | 0.16 | 3,060,741 | 0.05 | 340,082 |

Table 2 compares spend and token usage across seven models under a fixed nine-run budget. A run denotes one execution of a model on a fixed (task, difficulty) condition, during which the model generates $N$ samples; in Table 1, each row corresponds to one such run.

From the table, llama-3 is the most cost-efficient by a wide margin ($0.05/M tokens), followed by claude-opus-4 and grok-4 at $1.00/M and deepseek-r1 at $1.83/M; gpt-4o ($3.69/M), gpt-5 ($8.10/M), and gemini-2 ($8.69/M) are the most expensive per token, reflecting higher unit prices and longer average reasoning outputs. Despite low unit prices, deepseek-r1 and grok-4 accrued higher total spend due to large per-run budgets ($\sim$2.5M tokens/run), whereas llama-3 and claude-opus-4 remained inexpensive both per token and in total.

## B  EXTENDED RELATED WORK

**Benchmarks for Scientific Discovery with LLMs.**  Recent work explores LLMs for scientific reasoning and discovery via domain benchmarks and end-to-end "AI scientist" setups. Shojaee et al. introduce equation-discovery tasks across physics and chemistry (Shojaee et al., 2024; 2025), and Koblischke et al. propose Gravity for physics-inspired reasoning (Koblischke et al., 2025). Coignion et al. assess LLM code generation (Coignion et al., 2024), while Chen et al. develop Auto-Bench to evaluate full AI-scientist loops including query generation and experiment planning (Chen et al., 2025). These efforts provide valuable snapshots of scientific reasoning skills but predominantly emphasize single-answer correctness or task completion and typically assume one ground-truth solution per input.

**Creativity-Aware Benchmarks.**  Creativity has long been framed in psychology as producing outputs that are both *novel* and *appropriate*, with divergent thinking operationalized via *fluency*, *originality*, and *flexibility* (Guilford, 1950; Torrance, 1966; Amabile, 1996), and further conceptualized by Boden's taxonomy of combinational, exploratory, and transformational creativity (Boden, 2004). In computational creativity, formal criteria for evaluating novelty/value were articulated by Ritchie (Ritchie, 2007), and exploration-centric search (e.g., novelty search) was proposed to overcome objective-myopia (Lehman & Stanley, 2011). Building on this lineage, recent LLM benchmarks have begun to incorporate diversity/novelty. HypoBench (Liu et al., 2025) and IdeaBench (Guo et al., 2025b) quantify hypothesis novelty via instruction prompts and post-hoc analysis; CreativEval (DeLorenzo et al., 2024) measures fluency, flexibility, originality, and elaboration in LLM-generated code; LiveIdeaBench (Ruan et al., 2024) evaluates divergent thinking across Guilford's creativity dimensions; Nagarajan et al. design open-ended algorithmic tasks explicitly targeting originality and diversity, arguing next-token prediction is myopic for creative leaps and showing seed conditioning, multi-token objectives, and diffusion can elicit more diverse outputs (Nagarajan et al., 2025); and Wang et al. formalize Relative and Statistical Creativity as distributional indistinguishability from human creators, yielding practical measures for prompt-conditioned autoregressive models (Wang et al., 2024a). Beyond core NLP settings, Bhat et al. propose a domain-specific evaluation framework for marketing creativity with LLMs, operationalizing novelty and appropriateness using task-grounded criteria (Bhat et al., 2025).

However, most such evaluations rely on LLM-as-judge or human raters and lack a formally enumerated hypothesis space, making coverage and non-redundancy difficult to measure precisely. Our work complements these efforts by providing deterministic validators and exactly enumerated admissible sets, enabling model-agnostic, set-level measurement of Validity, Uniqueness, and Recovery.

## C   MODEL METADATA AND PROMPTS

Table 3: Models evaluated with provider and snapshot.

| Model | Provider | Version/Snapshot |
|---|---|---|
| GPT-5 | OpenAI | 2025-08-07 |
| Gemini-2.5-Pro | Google | 2025-06-17 |
| Claude-Opus-4 | Anthropic | 2025-05-22 |
| DeepSeek-R1 | DeepSeek | 2025-01-20 |
| Grok-4 | xAI | 2025-07-09 |
| GPT-4o | OpenAI | 2024-05-13 |
| LLaMA-3.3-70B-Instruct | Meta | 2024-12-06 |

Table 3 lists the models included in our evaluation and clarifies the access path. "Provider" refers to the model's origin organization (e.g., OpenAI, Google, Anthropic), not the API gateway; all models were accessed via OpenRouter except `GPT-5`, which was called through the OpenAI API. Full prompts and decoding settings are provided in our anonymous repository.

## D   USE OF LLMS

We used a large language model (ChatGPT) to aid polish wording (e.g., grammar, phrasing, figure captions). Besides, we also used Claude to refine the scripts, and checked by humans. We used ChatGPT for copy-editing (grammar, phrasing, figure captions) and Claude for script refinement. All code logic, experimental design, analyses, and claims were verified by the authors.

