# OpenReview forum: "HypoSpace: Evaluating LLM Creativity as Set-Valued Hypothesis Generators under Underdetermination"
_ICLR.cc/2026/Conference — Submitted to ICLR 2026_

### Official Review · Reviewer_1mJR · 2025-10-26

**Soundness:** 3
**Presentation:** 3
**Contribution:** 2
**Rating:** 2
**Confidence:** 4

**Summary:**

This paper studies the ability of LLMs to infer multiple hypotheses when fitting the observations of under-determined problems. The paper introduces 3 toy problems with exact enumeration of hypothesis space, which enables systematic evaluation of LLM solutions' validity, uniqueness, and recovery. Empirically, the authors find that frontier models often find valid solutions, but suffer mode collapse as the space of possible solutions increases.

While the motivation and paper are well written, I am leaning towards rejection because (1) the empirical findings are not novel, (2) the artificial nature of the problems limit the conclusions that can be made, (3) many of the reasoning LLMs are acing the first two tasks at all difficulty levels, limiting its discriminatory power.

1) The main experimental conclusion of the paper: "mode collapse of solutions in larger solution spaces" has been reported before. See [Duan et al., NeurIPS 2025](https://arxiv.org/abs/2507.02083)
2) The three problems are artificially constructed such that the solution space is known a priori. This seems very restrictive to me and fundamentally limits the generalization power of conclusions made via this diagnostic. It also seems misleading to me that the authors have chosen domain-specific names like "DNA interaction" and "3D understanding", when the actual problems are extreme simplifications of these complex problems.
3) Could the authors perhaps explore more difficult versions of the first two problems in Table 1? It seems too easy at the moment for the reasoning models.

**Strengths:**

- good motivation, clear writing for the most part
- measuring the divergent creativity of hypotheses from an information theory perspective is interesting
- experiments are sound and thorough

**Weaknesses:**

- originality of experimental findings is limited
- artificial nature of problems limits conclusions
- diagnostic lacks discriminatory power

**Questions:**

- Can the authors explain how they calculated the entropy gain in section 3.3? Are they counting the number of times the LLM repeated each solution to estimate the empirical frequency of that solution? If so, doesn't this defeat the purpose of using the actual posterior probability of each solution given the observations? I thought that $P(h_i | O)$ is a property of the problem itself, not of the LLM.
- Can the authors provide the prompt used to obtain solutions from the LLMs? Is the ability of finding diverse hypotheses confounded by the model's ability to follow prompts? Is the prompt explicitly encouraging diverse mode exploration?

---

> ### Author Response · Authors · 2025-11-21
>
> **W1.** We thank the reviewer for pointing us to this concurrent work. However, although Duan et al. (NeurIPS 2025) also use the term “mode collapse,” they study a different phenomenon from ours. Their result that model performance drops as system complexity increases only corresponds to what we measure as Validity Rate (VR), meaning a decline in correctness when the solution space becomes larger.
>
> In contrast, their paper does not evaluate the kind of mode collapse we focus on. They do not measure whether the generated hypotheses are mutually non-redundant, which is captured by our Uniqueness or Novelty Rate (NR). They also do not measure whether the model can recover the full set of admissible hypotheses, which is captured by our Recovery Rate (RR).
>
> Our notion of mode collapse refers to a loss of diversity in the hypothesis space, not just a decrease in accuracy. HypoSpace can detect this because it enumerates the full hypothesis space and checks whether the mode collapses to only a small subset of valid mechanisms even when many exist. This aspect is not addressed in Duan et al.
>
> We also note that Duan et al. is a very recent NeurIPS 2025 preprint released around the same time as our submission. We will add a brief clarification of these differences in the revised version.
>
>
> ---
>
> **W2.** Many thanks for the constructive comments. We intentionally make the solution space known a priori, since this is the only way to construct fully enumerable ground-truth spaces, which is essential for evaluating the diversity and coverage performance.
>
> However, this design choice does not restrict the generalization of the framework. In our work, each of task abstracts actual scientific problems:
> - **3D understanding** In real-world robot navigation, limited observations can correspond to multiple plausible spatial layouts, a well-known challenge in SLAM [R1]. Our 3D understanding task simulates this geometric underdetermination to evaluate whether existing LLMs can generate and enumerate all admissible scene layouts from limited views.
> - **Causal reasoning**  In cell-signaling [R2], edge directions are typically identified through targeted interventions. However, when interventions are limited, multiple causal structures remain equally compatible with the observed correlations. Our causal task abstracts this underdetermination and quantifies it using VR/NR/RR.
> - **Genetic interactions** In biological discovery [R3], limited phenotypic observations often admit multiple mechanistically distinct genetic‐interaction mechanisms; our Boolean task models this ambiguity directly.
>
> To further demonstrate, we conducted a complex real-world genetic interaction experiment. We used vesicle trafficking module from a real-world dataset [R3], which includes 6 genes and 14 knockout experiments (6 single, 8 double). To avoid data leakage, we anonymized the gene name as A-F. Given this anonymized dataset, the LLMs were tasked to propose hypotheses for cellular viability.
>
> As indicated in Table R1, performance on real-world data is comparable to synthetic benchmarks. Four state-of-the-art models (DeepSeek-R1, Gemini-2.5-Pro, GPT-5, Grok-4) achieved perfect performance with 100% VR and 70% NR, generating 7 unique valid hypotheses that converge to a consistent solution space, while three failed models (Claude Opus-4, GPT-4o, Llama-3.3-70b) demonstrated high diversity (80-100% NR) but 0% validity, indicating they explored extensively but in incorrect directions inconsistent with real biological observations.
>
>
> | Model | VR | NR | No. of Unique Valid Hypotheses |
> |-------|-----|--------|------------------------|
> | DeepSeek-R1 | 100% | 70% | 7 |
> | Gemini-2.5-Pro | 100% | 70% | 7 |
> | GPT-5 | 100% | 70% | 7 |
> | Grok-4 | 100% | 70% | 7 |
> | Claude Opus-4 | 0% | 100% | 0 |
> | GPT-4o | 0% | 100% | 0 |
> | Llama-3.3-70b | 0% | 80% | 0 |
>
> Table R1. Real-world genetic interaction experiment results.
>
>
> Regarding naming, we appreciate the reviewer’s point and we will revise task names to avoid any impression of overclaiming; they are meant to capture the type of ambiguity (e.g., interaction ambiguity, geometric ambiguity), not to model the full complexity of DNA biology or 3D vision. We will consistently use the names "3D Voxel Reconstruction" and "Boolean Genetic Interactions."
>
>
> [R1] Durrant-Whyte, H. and Bailey, T., 2006. Simultaneous localization and mapping: part I. IEEE robotics & automation magazine, 13(2), pp.99-110.
>
> [R2] Sachs, Karen, et al. "Causal protein-signaling networks derived from multiparameter single-cell data." Science 308.5721 (2005): 523-529.
>
> [R3] Costanzo M, et al. A global genetic interaction network maps a wiring diagram of cellular function. Science. 2016 Sep 23;353(6306):aaf1420. doi: 10.1126/science.aaf1420. PMID: 27708008; PMCID: PMC5661885.

---

> > ### Author Response · Authors · 2025-11-21
> >
> > **W3.** We appreciate the suggestion. While some models (e.g., GPT-5, Gemini-2.5-Pro, Grok-4) perform strongly on the first two tasks, others (e.g., Claude-Opus-4, DeepSeek-R1) still struggle, indicating that these tasks are not universally solved.
> >
> > Following the suggestion, we increase difficulties of causal task for three high-peformance models (GPT-5, Gemini-2.5-Pro, and Grok-4), the result is shown in Table R2 and R3. This high-VR, low-NR/RR pattern shows models understand domain constraints yet still collapse onto a narrow subset of admissible mechanisms.
> >
> >
> > | Model | VR | NR | RR |
> > |-------|--------------|-----------|---------------|
> > | GPT-5 | 100.0% (0% ↓)| 71.2% (-28.0% ↓) | 71.2% (-28.0% ↓) |
> > | Gemini 2.5 Pro | 100.0% (0% ↓)| 72.4% (-27.0% ↓) | 72.4% (-26.8% ↓) |
> > | Grok-4 | 100.0% (0% ↓)| 57.7% (-42.1% ↓) | 57.7% (-42.1% ↓)|
> >
> > Table R2. Performance of Causal Inference task across models (160 ground-truths). The bracket indicates the performance drop compared with the most difficult level of Causal Inference task in Table 1.
> >
> > | Model | VR | NR | RR |
> > |-------|--------------|-----------|---------------|
> > | GPT-5 | 60.8% (-39.2% ↓) | 24.8% (-74.0% ↓) | 23.2% (-74.0% ↓) |
> > | Gemini 2.5 Pro | 100.0% (0% ↓)| 64.8% (-30.4% ↓) | 64.8% (-30.3% ↓) |
> > | Grok-4 | 10.4% (-89.5% ↓)| 10.4% (-89.5% ↓) | 10.4% (-89.4% ↓)|
> >
> > Table R3. Performance of 3D Voxel Reconstruction task across models (125 ground-truths). The bracket indicates the performance drop compared with the most difficult level of 3D Voxel Reconstruction task in Table 1.
> >
> > ---
> >
> > **Q1.** Yes, we calculated entropy from the empirical frequencies of LLM-generated hypotheses (section 3.3 & Figure 3) instead of evaluating the posterior probability. The aim is to measure the model's exploration behaviour. The exploration entropy quantifies how broadly the model searches and therefore characterizes the model, not the task posterior. We understand that there might be confusions in our original manuscript, and we will revise the manuscript to clarify this.
> >
> > ---
> >
> > **Q2.** The prompts used for Causal Inference task are provided below:
> >
> >         You are given observations from perturbation experiments on a causal system.
> >
> >         Semantics:
> >         - When a node is perturbed, the perturbed node is 0.
> >         - A node is 1 if it is a downstream descendant of the perturbed node in the causal graph.
> >         - All other nodes are 0.
> >
> >         Nodes: {nodes_str}{constraint_info}
> >
> >         Observations:
> >         {obs_block}
> >
> >         Prior predictions (do not repeat if avoidable):
> >         {prior_block}
> >
> >         Task:
> >         Output a single directed acyclic graph (DAG) over the nodes above that explains all observations.
> >
> >         Diversity rule:
> >         - A "diverse" graph is any valid graph whose edge set is NOT identical to any prior prediction.
> >         - Generate diverse graphs when possible to explore the solution space.
> >
> >         Formatting rules:
> >         1) Use only the listed nodes. No self-loops. No cycles.
> >         2) Respond with exactly one line:
> >         - If there are edges: Graph: A->B, B->C
> >         - If there are no edges: Graph: No edges
> >
> > We do encourage the model to explore diverse hypotheses in the prompt. However, this does not confound our measurement: the prompt only asks the model to “try proposing different mechanisms,” but it does not specify how many or what kinds of hypotheses to generate. Thus, the level of diversity arises from the model’s own ability. HypoSpace simply measures how much non-redundant and valid diversity the model can actually produce.
> >
> > Moreover, HypoSpace uses history-conditioned rounds: at each step, the model receives the problem, observations, and all previously proposed mechanisms. This setup tests whether the model can avoid redundancy and continue expanding the hypothesis set on its own, beyond what is stated in the prompt.

---

> ### Comment · Reviewer_1mJR · 2025-11-26
>
> Thanks for the comprehensive rebuttal. I appreciate that the authors validated the models on a real world biological dataset. But my main concern remains that these highly abstracted problems are not tracking / measuring the performance of LLMs on the corresponding real-world problems that the authors claim. In a time today when public benchmarks are being saturated without us seeing truly impactful progress of LLMs in science related fields, I think that the utmost priority of novel benchmarks is to establish a clear relationship between progress on the benchmark and progress on the actual task of interest in the real world. For this reason, I will maintain my score.

---

### Official Review · Reviewer_Et1B · 2025-10-30

**Soundness:** 3
**Presentation:** 3
**Contribution:** 3
**Rating:** 4
**Confidence:** 2

**Summary:**

This paper introduces HypoSpace, a diagnostic suite designed to evaluate LLM creativity as the ability to generate sets of valid, non-redundant hypotheses under scientific underdetermination. Instead of giving score to single correct answers, it proposes to measure how well a model explores and covers all admissible explanations consistent with the same observations by defining three complementary metrics, including Validity (VR), Uniqueness (NR), and Recovery (RR). As experiments, the authors apply them to three designed tasks: causal graph inference, 3D voxel reconstruction, and Boolean genetic interaction. Empirical experiments across frontier LLMs such as GPT-5 reveal that high validity but strong degradation in uniqueness and recovery as the hypothesis space enlarges, reflecting the (so-called) mode collapse in reasoning.

**Strengths:**

- Empirical results revealed some insight regarding the “mode collapse phenomenon” in reasoning models with the proposed VR/NR/RR that traditional accuracy-based metrics cannot capture.
- Proposed three relevant diagnostic tasks including causal graph inference, 3D voxel reconstruction, and Boolean genetic interaction, to study the capability of LLMs for sampling hypothesis. If these are open-sourced, it can help the analysis of LLMs in this regard for future research.

**Weaknesses:**

- The evaluation tasks are relatively toy and over-simplified. Being far from realistic scientific inference scenario can weaken the contribution of this study. One case study of real-world problem can help.
- The proposed indicators can serve as sufficient measure for simple hypothesis generation cases like the used three tasks, but may probably not be qualified for complex problem where scientific hypothesis can be tricky to evaluate with them (in my opinion).

**Questions:**

- Does the degradation in uniqueness and recovery measures is caused simply by the increase of “difficulty” of commonly studied reasoning problem? Can you explain what are the difference between increased hypothesis space size and increased difficulty in the context of this paper?
- When the |H_o| grows, does the corresponding change of sampling hyperparameters like temperature and top-p value helps improve the validity or uniqueness metrics? Since the problem scales up, I assume the model may need to re-balance the exploration somehow.
- Corresponding to one of the concerns above, can you also show some real-world or open-ended problems of generating hypothesis? This may make the proposed diagnostic framework, HypoSpace, more convincing and useful.
- What strategies do you think will ameliorate the collapse observed from the results? Via training-free prompting or proper finetuning?

---

> ### Author Response · Authors · 2025-11-21
>
> **W1 & Q3.**
> Thank you for raising this insightful question. We would like to clarify that our tasks are motivated from real scientific problems.
> - **3D understanding** In real-world robot navigation, limited observations can correspond to multiple plausible spatial layouts, a well-known challenge in SLAM [R1]. Our 3D understanding task simulates this geometric underdetermination to evaluate whether existing LLMs can generate and enumerate all admissible scene layouts from limited views.
> - **Causal reasoning**  In cell-signaling [R2], edge directions are typically identified through targeted interventions. However, when interventions are limited, multiple causal structures remain equally compatible with the observed correlations. Our causal task abstracts this underdetermination and quantifies it using VR/NR/RR.
> - **Genetic interactions** In biological discovery [R3], limited phenotypic observations often admit multiple mechanistically distinct genetic‐interaction mechanisms; our Boolean task models this ambiguity directly.
>
>
> Indeed, these evaluation tasks are intentionally simplified because this is the only way to construct fully enumerable ground-truth spaces. With such enumerable ground-truth spaces, we can precisely measure a model’s validity, uniqueness, and coverage, allowing us to evaluate how the model generates and expands the set of admissible explanations without ambiguity.
>
> Following the reviewer’s suggestion, we conducted a complex real-world genetic interaction experiment. We used vesicle trafficking module from a real-world dataset [R3], which includes 6 genes and 14 knockout experiments (6 single, 8 double). To avoid data leakage, we anonymized the gene name as A-F. Given this anonymized dataset, the LLMs were tasked with proposing hypotheses for cellular viability.
>
> As indicated in Table R1, the real-world results yield the same overall conclusion as our abstracted datasets. Four state-of-the-art models (DeepSeek-R1, Gemini-2.5-Pro, GPT-5, Grok-4) achieved perfect performance with 100% VR and 70% NR, generating 7 unique valid hypotheses that converge to a consistent solution space, while three failed models (Claude Opus-4, GPT-4o, Llama-3.3-70b) demonstrated high diversity (80-100% NR) but 0% validity, showing broad exploration but none of the generated hypotheses were biologically valid.
>
>
> | Model | VR | NR | No. of Unique Valid Hypotheses |
> |-------|-----|--------|------------------------|
> | DeepSeek-R1 | 100% | 70% | 7 |
> | Gemini-2.5-Pro | 100% | 70% | 7 |
> | GPT-5 | 100% | 70% | 7 |
> | Grok-4 | 100% | 70% | 7 |
> | Claude Opus-4 | 0% | 100% | 0 |
> | GPT-4o | 0% | 100% | 0 |
> | Llama-3.3-70b | 0% | 80% | 0 |
>
> Table R1. Real-world genetic interaction experiment results.
>
> [R1] Durrant-Whyte, H. and Bailey, T., 2006. Simultaneous localization and mapping: part I. IEEE robotics & automation magazine, 13(2), pp.99-110.
>
> [R2] Sachs, Karen, et al. "Causal protein-signaling networks derived from multiparameter single-cell data." Science 308.5721 (2005): 523-529.
>
> [R3] Costanzo M, et al. A global genetic interaction network maps a wiring diagram of cellular function. Science. 2016 Sep 23;353(6306):aaf1420. doi: 10.1126/science.aaf1420. PMID: 27708008; PMCID: PMC5661885.
>
> ---
>
> **W2.** We agree that HypoSpace does not cover more complex scientific cases whose hypothesis spaces are not fully enumerable, since such settings make validity and coverage impossible to measure exactly.
>
> However, to answer the reviewer’s concern, complex scientific reasoning problems can be studied by simplifying them into small, exactly solvable instances. For example, large-scale Ising models with thousands of spins are routinely analyzed using 2×2 or 3×3 lattices that preserve the essential structure while remaining fully enumerable [R4].
>
> Based on this principle, we intentionally abstract several real-world scenarios into fully enumerable tasks. This design allows us to precisely measure validity, uniqueness, and coverage, and therefore assess how the model generates and expands sets of admissible explanations without ambiguity.
>
>
> [R4] Lucas, A. “Ising formulations of many NP problems.” Frontiers in Physics, 2:5, 2014.

---

> > ### Author Response · Authors · 2025-11-21
> >
> > **Q1.** In our setting, “difficulty” of commonly studied reasoning problems corresponds to what we measure as the Validity Rate (VR), which reflects only whether the LLM produces correct hypotheses. Even if all outputs collapse to the same correct explanation, VR can still reach 100%.
> >
> > In contrast, the Uniqueness/Novelty Rate (NR) evaluates whether the generated hypotheses are mutually non-redundant, and the Recovery Rate (RR) evaluates whether the model can recover the entire set of admissible hypotheses. These metrics capture exploration rather than correctness, and therefore are not tied to the traditional notion of task difficulty.
> >
> > Regarding hypothesis space size, increasing the hypothesis space does not necessarily reduce VR, since a model may still output one or two correct hypotheses. As shown in Table 1, GPT-5 and Gemini-2.5-Pro maintain 100% VR even as the hypothesis space grows in the Causal Inference task.
> >
> > However, NR and RR become substantially harder to achieve, because generating many distinct and admissible hypotheses without redundancy is increasingly challenging. This is what we refer to as diversity difficulty, which is conceptually different from standard reasoning difficulty.
> >
> > ---
> >
> > **Q2.** We did not adjust temperature or top-p as $|H_O|$ grows. To ensure fairness and comparability across settings, we fixed all sampling hyperparameters for every model and every difficulty level.
> >
> > Our goal is to evaluate the model’s intrinsic exploratory ability, not diversity induced by sampling noise. Allowing temperature/top-p to vary with $|H_O|$ would make results incomparable across tasks and would mix model capability with hyperparameter tuning.
> >
> > ---
> >
> > **Q4.** Proper finetuning may help mitigate the observed collapse in diversity. For instance, incorporating diversity-aware objectives (e.g., rewards or losses that penalize redundant hypotheses) could encourage the model to explore a broader range of admissible and valid mechanisms rather than repeatedly generating a narrow set of modes.

---

> > > ### Comment · Reviewer_Et1B · 2025-11-26
> > >
> > > Thanks for the author for answering the questions. I would maintain my scoring.

---

### Official Review · Reviewer_3qvx · 2025-10-31

**Soundness:** 3
**Presentation:** 4
**Contribution:** 3
**Rating:** 6
**Confidence:** 4

**Summary:**

Utilizing LLM's potential for scientific discovery is an exciting and challenging problem that has been a focus recently. However this work proposes a clean and effective way of identifying how good current models are in proposing supporting hypothesis for the observed data. I like the framework as it provides a holistic way of evaluating for this task, which goes beyond identifying the accuracy of the hypothesis which prior works optimise for. By evaluating validity, uniqueness and recovery, the framework identifies if models can act as strong samplers of all possible hypothesis which can be helpful for scientific discovery. Unlike most creativity or hypothesis-generation benchmarks that depend on human or LLM-as-judge scoring, HypoSpace uses deterministic validators and exactly enumerated admissible sets, removing subjectivity and enabling reproducible, model-agnostic evaluation

**Strengths:**

Overall I believe this work is strong, some of its strengths are:

1. I like the tasks this work targeted, they seem novel and widely different from typical applications past work evaluate on.

2. This combination of ideas from psychology’s notion of divergent thinking with a concrete, enumerated approach to evaluating hypotheses feels genuinely fresh and thoughtfully put together.

3. Rigorous empirical evaluation of current frontier models is useful for understanding the current state of LLMs for this task

4. The modularity of the framework makes it directly applicable to other tasks, improving its potential generalization and usability in real world.

**Weaknesses:**

There are some aspects the work can improve upon:

1. While the paper quantifies exploration failure nicely, it doesn’t deeply analyze why models prefer certain hypotheses or how internal reasoning traces differ between models.

2. I am curious and confused about one thing: is it necessary for an agent to identify all possible solutions which justify the given observation data? I believe validating the number of potential hypothesis a model can come up with makes sense for certain tasks (more specifically when the subjectivity of the task is high), while identifying a smaller subset or getting the correct answer might be of higher importance in other cases.

**Questions:**

1. It would be interesting to see in how many cases the subset of hypothesis identified by the model were the most optimal solutions, or in how many cases the subset of defined hypothesis did not contain the top 3 or top 5 optimal solutions? In causal graph example, an optimal solution could potentially be given the nodes, which graphs require the smallest connections to justify the given observations?

2. I’m curious about why certain hypotheses are favored more frequently than others. It might be interesting to analyze which samples are easier for models to estimate and which are harder, and then compare their coverage. Such an analysis could guide targeted training to improve how LLMs explore hypothesis spaces. For instance, in the causal inference task, some graphs might only require adding a single edge to satisfy the interventional observations, while others may need multiple structural changes. If models consistently prefer these “easier” hypotheses, it would point to a deeper limitation in their ability to explore more complex or less accessible regions of the hypothesis space.

3. I am also curious if there is a way to identify if the model is actually equipped enough in identifying all possible hypothesis, or is it not good at this task. For example this is the first work I have seen where LLM is evaluated on 3D pixel reconstruction, is it possible that the model is just bad at this task, and therefore evaluating its ability to generate all possible hypothesis will be limited? It would be impactful to find simple cases where the model tends to have a high performance in getting the correct answer, but fails on uniqueness and recovery metrics.

I am happy to raise my scores, if the authors can provide some answers for the questions

---

> ### Author Response · Authors · 2025-11-21
>
> **W1. Why models prefer certain hypotheses; how traces differ.**
> Following the suggestion, we conducted a follow-up analysis of chain-of-thought reasoning traces on the Boolean task, where all models exhibit catastrophic failure (0% recovery on complex expressions). We selected severely underdetermined set (1 observation, 40 ground truths) and prompted 3 models (GPT-4o, Grok4, Gemini-2.5-Pro) to generate hypotheses with explicit reasoning ("Think step-by-step and explain your reasoning for each hypothesis you consider"). We analyzed three dimensions via automated text processing:
>
> - **Simplicity bias**: Lexical pattern matching counting mentions of `simple`, `basic`, `straightforward`, `most likely`, `enough to`, `should be sufficient` in the reasoning section
> - **Termination patterns**: Keyword detection classifying why models stopped:
>    - **Sufficiency**: `sufficient`, `should cover`,`thorough sample` ,`infinitely long`，`unbounded`, etc (claims of adequacy).
>    - **Exhaustion**: `all possible`, `exhausted`, `exhaustive`, `covered all`, etc (claim of completeness).
>
> The result is shown in Table R1. We can find that:
> - All models explicitly prioritize low-complexity expressions. This manifests as breadth-first search by operator count, exhausting 1-3 operator space but under-sampling 4-5 operators where most valid hypotheses reside.
> - The three models exhibit distinct stopping patterns:
>     - GPT-4o claims "systematically generated through consideration of expressions with increasing complexity" despite covering only 9/40.
>     - Grok-4 states "thorough but practical, as further expansion would be unbounded", yet only 22/40 valid ≤5-operator expressions found within its stated budget.
>     - Gemini-2.5-Pro gives prompt like "thorough sample of expressions up to 5 operators, as the list is infinitely long"—but 40 are enumerable and 17 are missing.
>     All three exhibit **false convergence**: terminating based on subjective sufficiency signals despite objective incompleteness.
>
> | Model | Simplicity Bias | Termination | Generated | Coverage |
> |-------|-----------------|-------------|-----------|----------|
> | Gemini-2.5-Pro | 3 | Sufficiency | 23/40 | 57.5% |
> | GPT-4o | 3 | Exhaustion | 9/40 | 22.5% |
> | Grok-4 | 3 | Sufficiency | 22/40 | 55.0% |
>
> Table R1. Reasoning analysis across models.

---

> > ### Author Response · Authors · 2025-11-21
> >
> > **W2. Thank you for the thoughtful question. We would like to address it point-by-point, and we will incorporate the corresponding clarifications in the revised manuscript.**
> >
> > **(1) Why we identify all possible solutions?**
> > Thanks for pointing this out. We agree that identifying a smaller subset at some cases is sufficient. However, enumerating all admissible explanations remains essential because this strategy allows us to accurately measure the diversity and coverage of the ground-truth space. Our HypoSpace targets a specific but important class of problems: underdetermined inference, where observations alone do not distinguish among multiple mechanistically distinct hypotheses, making it impossible to identify "the correct answer" without additional experiments. For example, one problem might have multiple equally plausible explanations; as Spirtes et al. [R1] note, “If two causal structures can equally account for the same statistics, then no statistics can distinguish them.” In these settings, enumerating the set of admissible hypotheses (or a representative cover) is a prerequisite for designing discriminative experiments that enable convergence.
> >
> > **(2) When is enumerating all (or most) necessary?**
> > We give several examples where enuerating all is necessary:
> > - In real-world robot navigation, limited observations can correspond to multiple plausible spatial layouts, a well-known challenge in SLAM [R3]. Our 3D understanding task simulates this geometric underdetermination to evaluate whether existing LLMs can generate and enumerate all admissible scene layouts from limited views.
> > - In cell-signaling [R4], edge directions are typically identified through targeted interventions. However, when interventions are limited, multiple causal structures remain equally compatible with the observed correlations. Our causal task abstracts this underdetermination and quantifies it using VR/NR/RR.
> > - In biological discovery [R2], limited phenotypic observations often admit multiple mechanistically distinct genetic‐interaction mechanisms; our Boolean task models this ambiguity directly.
> >
> > **(3) When is a smaller subset sufficient?**
> > The conditions under which a smaller subset of hypotheses is sufficient include cases where the problem is well-determined with a clear ground truth (e.g., factual QA or standard classification), where a single identifiable optimum exists (as in many optimization tasks), or where diversity is subjective rather than mechanistic.
> >
> > **(4) Our contribution:**
> > HypoSpace measures whether a model can systematically explore underdetermined hypothesis spaces or if it exhibits mode collapse (high VR but low RR). This capability matters even if not all applications require it. To evaluate this, we abstract three real-world scientific problems into simulated tasks with fully enumerated hypothesis spaces as described in (2). This enables precise measurement of a model’s diversity and coverage. We further analyze how current LLMs, including both “thinking” and non-“thinking” models, perform under underdetermination, and find that they still have substantial room for improvement in exploration and creativity.
> >
> > [R1] Spirtes, P., Glymour, C. and Scheines, R., Causation, Prediction, and Search, 2nd ed. Cambridge, MA: MIT Press, 2000.
> >
> > [R2] Costanzo M, et al. A global genetic interaction network maps a wiring diagram of cellular function. Science. 2016 Sep 23;353(6306):aaf1420. doi: 10.1126/science.aaf1420. PMID: 27708008; PMCID: PMC5661885.
> >
> > [R3] Durrant-Whyte, H. and Bailey, T., 2006. Simultaneous localization and mapping: part I. IEEE robotics & automation magazine, 13(2), pp.99-110.
> >
> > [R4] Sachs, Karen, et al. "Causal protein-signaling networks derived from multiparameter single-cell data." Science 308.5721 (2005): 523-529.

---

> > > ### Author Response · Authors · 2025-11-21
> > >
> > > **Q1&Q2** We appreciate the suggestion. In underdetermined inference, there is no task-intrinsic optimal solution without an external criterion. By definition, underdetermination means observations are consistent with multiple mechanistically distinct hypotheses, and no unique optimum can be identified from observations alone. For example, in Boolean genetic interaction task, given observations, both "y" and "x OR y" may be equally consistent. Which is "optimal"? Without additional criteria, neither is objectively better. If one could identify "top-k optimal solutions," the problem would be well-determined, not underdetermined.
> > >
> > > However, we can still examine whether models exhibit systematic preferences (e.g., for simpler/sparser hypotheses). In order to fill in the gap, we add these experiments:
> > >
> > > We firstly define complexity based on task-specific structural features:
> > > - Boolean: Simple (≤3 operators) vs. Complex (4-5 operators)
> > > - Causal: Simple (≤3 edges) vs. Complex (>3 edges)
> > > - 3D: Simple (≥50% ground bias*) vs. Complex (<50% ground bias)
> > >
> > > *Ground bias = proportion of blocks on the ground layer/Total number of blocks palced.*
> > >
> > > Then the following metrics are analyzed:
> > > - Generation Rate (Less Complex/Complex): It measures the distribution of complexity among all valid hypotheses generated by the model (including duplicates, excluding formatting errors), revealing the model's exploration bias during generation
> > > - Recovery Rate (RR Less / RR Complex): It measures the proportion of ground truth hypotheses successfully recovered by the model, separately for simple and complex GTs, quantifying the model's actual capability to recover hypotheses of each complexity level.
> > > - Recovery Gap: RR Less - RR Complex
> > > - Overall RR & NR (aligned with Table 1).
> > >
> > > As indicated in Table R2, we observe three consistent phenomena:
> > > - **Bias toward less complex hypotheses.** Models over-generate low-complexity forms: e.g., GPT-4o 93%/7% (simple/complex) in causal; in Boolean (Hard) most models are ~100%/0%.
> > > - **Higher recovery on less-complex sets.** RR_Less > RR_Complex across weaker/mid models (e.g., Claude-Opus-4 causal 71% vs 36%, +35% gap), consistent with preferring hypotheses that require minimal edits to fit observations.
> > > - **Collapse on complex regions.** In Boolean (Hard), RR_Complex = 0% for all models, showing systematic avoidance of deeper programs even when VR is high.
> > >
> > > | Model | Generation Rate<br>(L / C) | RR Less | RR Complex | RR Gap | Overall RR | NR |
> > > |-------|---------------------|---------|------------|--------|------------|-----|
> > > | **Causal (n6, ≤3 / >3 edges)** | | | | | | |
> > > | GPT-4o | 93% / 7% | 11% | 5% | +6% | 7% | 23% |
> > > | LLaMA-3 | 91% / 9% | 4% | 1% | +3% | 2% | 38% |
> > > | GPT-5 | 47% / 53% | 100% | 99% | +1% | 99% | 99% |
> > > | Claude-Opus-4 | 67% / 33% | 71% | 36% | +35% | 51% | 90% |
> > > | Grok-4 | 47% / 53% | 100% | 100% | +0% | 100% | 100% |
> > > | Gemini-2.5-Pro | 47% / 53% | 100% | 98% | +2% | 99% | 99% |
> > > | DeepSeek-R1 | 54% / 46% | 96% | 66% | +30% | 79% | 81% |
> > > | **3D (dim 3×3×3, ≥50% / <50% GB)** | | | | | | |
> > > | GPT-4o | 92% / 8% | 13% | 2% | +11% | 14% | 58% |
> > > | LLaMA-3 | 92% / 8% | 5% | 0% | +5% | 6% | 67% |
> > > | GPT-5 | 64% / 36% | 100% | 97% | +3% | 99% | 99% |
> > > | Claude-Opus-4 | 83% / 17% | 54% | 20% | +35% | 49% | 81% |
> > > | Grok-4 | 63% / 37% | 100% | 99% | +1% | 100% | 100% |
> > > | Gemini-2.5-Pro | 67% / 33% | 100% | 87% | +13% | 95% | 95% |
> > > | DeepSeek-R1 | 70% / 30% | 91% | 73% | +18% | 85% | 86% |
> > > | **Boolean (Hard, ≤3 / 4-5 ops)** | | | | | | |
> > > | GPT-4o | 100% / 0% | 21% | 0% | +21% | 14% | 21% |
> > > | LLaMA-3 | 100% / 0% | 17% | 0% | +17% | 11% | 14% |
> > > | GPT-5 | 98% / 2% | 68% | 0% | +68% | 48% | 50% |
> > > | Claude-Opus-4 | 98% / 2% | 35% | 0% | +35% | 24% | 24% |
> > > | Grok-4 | 96% / 4% | 52% | 0% | +52% | 36% | 38% |
> > > | Gemini-2.5-Pro | 98% / 2% | 67% | 0% | +67% | 47% | 49% |
> > > | DeepSeek-R1 | 97% / 3% | 53% | 0% | +53% | 36% | 37% |
> > >
> > > Table R2. Model Preferences for Three Tasks
> > >
> > > ---
> > >
> > > **Q3.** Thanks for pointing this out. We have introduced the Validity Rate (VR) to measure whether an LLM has sufficient task-specific knowledge. A high VR simply indicates that the model possesses the necessary domain knowledge, because it can provide correct answers even if those answers are identical. As shown in Table 1, in the 3D Voxel Reconstruction task, GPT-5 achieves a VR of 100%.
> > >
> > > Therefore, our conclusions driven from the three tasks still hold: almost all thinking-based LLMs exhibit very high VR, indicating that they do possess the required task-specific knowledge. Their low NR and RR scores demonstrate that the models suffer from diversity or coverage collapse, rather than a lack of domain knowledge.

---

### Official Review · Reviewer_oycp · 2025-11-03

**Soundness:** 3
**Presentation:** 3
**Contribution:** 2
**Rating:** 2
**Confidence:** 4

**Summary:**

This paper introduces HypoSpace, a framework for evaluating large language models as set-valued hypothesis generators in underdetermined reasoning settings—where multiple valid explanations exist for the same evidence. Rather than focusing solely on accuracy, HypoSpace quantifies three dimensions of model performance: Validity, measuring the proportion of correct hypotheses; Uniqueness, capturing the diversity of generated solutions; and Recovery, reflecting how completely the model explores the admissible hypothesis space. The benchmark spans three structured domains—causal graph inference, 3D voxel reconstruction, and Boolean genetic interaction modeling.

**Strengths:**

The framework is methodically clean, deterministic, and reproducible. It formalizes a relatively underexplored dimension of LLM evaluation—output diversity—in a controlled setting. The domains span causal, spatial, and symbolic reasoning, and the inclusion of exact validators is elegant from a benchmarking standpoint. The analysis is thorough, and results are clearly visualized. From an engineering perspective, the pipeline for enumerating hypothesis spaces and evaluating them under canonicalization is well executed.

**Weaknesses:**

The paper’s core motivation is underdeveloped: it assumes that producing multiple distinct hypotheses is intrinsically valuable without explaining when or why that matters for reasoning or creativity. In most real reasoning contexts, the goal is not to enumerate all possible explanations, but to converge efficiently on the most plausible or useful one. Thus, the emphasis on “recovery” and “uniqueness” risks being arbitrary rather than insightful.
The evaluation tasks are artificial, limited to small, fully enumerable spaces that have little connection to real-world cognitive or scientific problems. High validity scores in the causal and 3D tasks indicate that the framework measures diversity in trivial domains, not meaningful reasoning skill.
The difficulty scaling is not adaptive; as task complexity grows, models degrade simply because they are not tuned for those domains, not because they lack creative capacity. This confounds interpretability of the NR/RR decline.
Finally, the claim of novelty is overstated: evaluating exploration in LLMs has been explored through uncertainty estimation, ensemble sampling, and information-theoretic diversity measures. HypoSpace provides a clean but limited instantiation of an old idea, not a new paradigm.

**Questions:**

Why is generating multiple hypotheses preferable or necessary compared to generating a single, well-reasoned explanation?

How do VR, NR, and RR translate to practical reasoning competence in real domains such as scientific discovery, design, or causal inference at scale?

If causal and voxel domains saturate easily, do these results truly measure reasoning or simply expose ceiling effects in trivial environments?

How would the authors calibrate difficulty relative to model capacity to avoid conflating lack of domain tuning with reasoning limits?

Could similar insights be achieved through simpler sampling-based diversity metrics rather than elaborate enumerated hypothesis spaces?

---

> ### Author Response · Authors · 2025-11-21
>
> **Weakness && Q1. Importance of generating multiple hypotheses:** We agree that identifying the most plausible hypothesis is important. However, enumerating all admissible explanations remains essential, because different explanations can be equally consistent with the observations at a given moment. For example, one problem might have multiple equally plausible explanations; as Spirtes et al. [R1] note, “If two causal structures can equally account for the same statistics, then no statistics can distinguish them.”
>
> [R1] Spirtes, P., Glymour, C. and Scheines, R., Causation, Prediction, and Search, 2nd ed. Cambridge, MA: MIT Press, 2000.
>
> --- --
>
> **Weakness. (1) Why fully enumerable tasks:** The intention of fully enumerable tasks is to obtain precise, unambiguous diagnostics of hypothesis-set generation. If the ground-truth space is not enumerable, benchmarking becomes unreliable because coverage of the generated hypotheses cannot be objectively assessed.
>
> **(2) Connection to real scientific problems:** Each of task abstracts a real-world scientific scenario where multiple explanations coexist:
>
> - **3D understanding** In real-world robot navigation, limited observations can correspond to multiple plausible spatial layouts, a well-known challenge in SLAM [R2]. Our 3D understanding task simulates this geometric underdetermination to evaluate whether existing LLMs can generate and enumerate all admissible scene layouts from limited views.
> - **Causal reasoning**  In cell-signaling [R3], edge directions are typically identified through targeted interventions. However, when interventions are limited, multiple causal structures remain equally compatible with the observed correlations. Our causal task abstracts this underdetermination and quantifies it using VR/NR/RR.
> - **Genetic interactions** In biological discovery [R4], limited phenotypic observations often admit multiple mechanistically distinct genetic‐interaction mechanisms; our Boolean task models this ambiguity directly.
>
>
> To further demonstrate, we conducted a complex real-world genetic interaction experiment. We used vesicle trafficking module from a real-world dataset [R4], which includes 6 genes and 14 knockout experiments (6 single, 8 double). To avoid data leakage, we anonymized the gene name as A-F. Given this anonymized dataset, the LLMs were tasked to propose hypotheses for cellular viability.
>
> As indicated in Table R1, performance on real-world data is comparable to synthetic benchmarks. Four SOTA models achieved perfect performance with 100% VR and 70% NR, generating 7 unique valid hypotheses that converge to a consistent solution space, while three failed models demonstrated high diversity (80-100% NR) but 0% validity, indicating they explored extensively but in incorrect directions inconsistent with real biological observations.
>
>
> | Model | VR | NR | No. of Unique Valid Hypotheses |
> |-------|-----|--------|------------------------|
> | DeepSeek-R1 | 100% | 70% | 7 |
> | Gemini-2.5-Pro | 100% | 70% | 7 |
> | GPT-5 | 100% | 70% | 7 |
> | Grok-4 | 100% | 70% | 7 |
> | Claude Opus-4 | 0% | 100% | 0 |
> | GPT-4o | 0% | 100% | 0 |
> | Llama-3.3-70b | 0% | 80% | 0 |
>
> Table R1. Real-world genetic interaction experiment results.
>
>
> [R2] Durrant-Whyte, H. and Bailey, T., 2006. Simultaneous localization and mapping: part I. IEEE robotics & automation magazine, 13(2), pp.99-110.
>
> [R3] Sachs, Karen, et al. "Causal protein-signaling networks derived from multiparameter single-cell data." Science 308.5721 (2005): 523-529.
>
> [R4] Costanzo M, et al. A global genetic interaction network maps a wiring diagram of cellular function. Science. 2016 Sep 23;353(6306):aaf1420. doi: 10.1126/science.aaf1420. PMID: 27708008; PMCID: PMC5661885.
>
> ---
>
> **Q2. Translation of VR, NR, and RR to practical competence.** Our metrics are designed to measure complementary abilities of hypothesis generation that matter in real scientific workflows:
> - VR (constraint satisfaction) measures the correctness of the generated hypotheses. Eg., in drug design, a candidate must satisfy known activity or pharmacological constraints before synthesis or testing.
> - NR (distinctness of ideas) measures non-redundancy across all proposed hypotheses, reflecting the model’s ability to diversify its ideas. Eg., in iterative scientific workflows, repeated hypotheses provide no new information.
> - RR (breadth under underdetermination) quantifies coverage of the admissible hypothesis space (correct and non-redundant), which is essential in tasks such as experimental design. Eg., in causal discovery, covering the Markov equivalence class reveals which interventions will most efficiently disambiguate competing graphs.
>
> Together, VR assesses the correctness of generated hypotheses, while NR and RR evaluate divergent exploration efficiency. At scale, higher NR/RR lead to fewer redundant experiments and fewer missed regions of the solution space under the same budget—directly improving practical discovery.

---

> > ### Author Response · Authors · 2025-11-21
> >
> > **Weakness && Q3**. Validity Rate (VR) does not quantify diversity. It only measures whether generated hypotheses are consistent with observations. High VR simply indicates that models possess the necessary domain knowledge; if all answers are correct but identical, VR would still be 100%.
> >
> > In contrast, our Uniqueness/Novelty Rate (NR) evaluates whether hypotheses are mutually non-redundant, and our Recovery Rate (RR) measures coverage over the admissible hypothesis space. These metrics directly capture exploration and creativity rather than correctness alone.
> >
> > Notably, the causal and 3D domains do not saturate uniformly across models: some models achieve high RR while others perform poorly, showing that these tasks do not produce ceiling effects but reveal meaningful differences in exploration ability. We increase difficulties of causal task for three high-peformance models (GPT-5, Gemini-2.5-Pro, and Grok-4), the result is shown in Table R2 and R3. This high-VR, low-NR/RR pattern shows models understand domain constraints yet still collapse onto a narrow subset of admissible mechanisms（i.e., no ceiling effect.）
> >
> > | Model | VR | NR | RR |
> > |-------|--------------|-----------|---------------|
> > | GPT-5 | 100.0% (0% ↓)| 71.2% (-28.0% ↓) | 71.2% (-28.0% ↓) |
> > | Gemini 2.5 Pro | 100.0% (0% ↓)| 72.4% (-27.0% ↓) | 72.4% (-26.8% ↓) |
> > | Grok-4 | 100.0% (0% ↓)| 57.7% (-42.1% ↓) | 57.7% (-42.1% ↓)|
> >
> > Table R2. Performance of Causal Inference task across models (160 ground-truths). The bracket indicates the performance drop compared with the most difficult level of Causal Inference task in Table 1.
> >
> > | Model | VR | NR | RR |
> > |-------|--------------|-----------|---------------|
> > | GPT-5 | 60.8% (-39.2% ↓) | 24.8% (-74.0% ↓) | 23.2% (-74.0% ↓) |
> > | Gemini 2.5 Pro | 100.0% (0% ↓)| 64.8% (-30.4% ↓) | 64.8% (-30.3% ↓) |
> > | Grok-4 | 10.4% (-89.5% ↓)| 10.4% (-89.5% ↓) | 10.4% (-89.4% ↓)|
> >
> > Table R3. Performance of 3D Voxel Reconstruction task across models (125 ground-truths). The bracket indicates the performance drop compared with the most difficult level of 3D Voxel Reconstruction task in Table 1.
> >
> > ---
> >
> > **Weakness && Q4**
> > Our Validity Rate (VR) measures the proportion of hypotheses that are consistent with the observations. A high VR indicates that the model understands the task-specific domain constraints and can generate valid hypotheses. As shown in Table 1 and R1, even in the hardest settings of our Causal Inference and 3D Understanding tasks, most reasoning-based LLMs still achieve very high VR, indicating that domain knowledge is not the limiting factor.
> >
> > Our difficulty scaling is therefore reasonable and not driven by domain mismatch: models maintain near-perfect VR even as difficulty increases, showing that they understand the domain across all levels. What drops is NR/RR, not VR, which isolates the degradation to exploration and reasoning limits rather than domain tuning.
> >
> > ---
> >
> > **Weakness. Differences between our work and others**
> >
> > **Uncertainty estimation** looks at how confident a model is in its answers. However, confidence does not tell us how creative the model is, or whether it has found all the possible valid answers.
> >
> > **Ensemble sampling** aggregates multiple model outputs or sampling trajectories to improve robustness and surface-level diversity. However, this strategy cannot check whether the answers are truly correct (VR), nor whether the outputs cover all valid solutions (RR) or unique mechanisms (NR).
> >
> > **Information-theoretic** diversity measures how spread out the model’s output distribution is. However, without knowing the valid hypothesis space, it can only approximate uniqueness (NR) and cannot accurately measure whether the diversity is meaningful or valid (VR & RR).
> >
> >
> > In contrast, HypoSpace introduces enumerable hypothesis spaces that enable accurate evaluation of VR, NR, and RR. This allows us to cleanly separate validity from exploration and to identify a failure mode (high VR but low RR) that prior methods cannot capture. Taken together, accurate definition and evaluation of creativity has not been available in prior work, positioning HypoSpace as a new diagnostic framework rather than a variant of existing methods.
> >
> > ---
> >
> > **Q5.** Sampling-based evaluation typically utilizes multiple stochastic generations to measure the dispersion of outputs. However, without enumerating the hypothesis space, these evaluations can only approximate uniqueness (NR) to some extent and cannot accurately assess the true diversity of outputs (VR & RR).
> >
> > Instead, HypoSpace differs by using fully enumerable hypothesis spaces, allowing us to compare a model’s generated set against the complete admissible set. This enables objective assessment of whether the model meaningfully expands its hypotheses rather than merely sampling variations. Sampling alone cannot provide this level of interpretability.

---

### Author Response · Authors · 2025-12-03
**General Response**

**General Response**
---
We appreciate the reviewers’ detailed feedback. We are encouraged by the broad recognition of HypoSpace's core strengths:

**Novelty and Motivation:** Well-motivated work proposing a novel (Reviewer 3qvx) /interesting idea (Reviewer 1mJR), which formalizes output diversity (Reviewer oycp).

**Methodology and Experiment:** Exact experiments and measurements achieved by utilizing enumerating hypothesis spaces and including exact validators (Reviewer oycp, 3qvx, Et1B).

**Empirical Strength:** The analysis is thorough, and results are clearly visualized. Experiments are sound (Reviewer oycp, 3qvx, 1mJR).

**Generalization:** The framework has good generalization capabilities due to its modularity (Reviewer 3qvx, Et1B).


**Summary of Rebuttal**
---

**Importance of generating multiple hypotheses (->Reviewer oycp, 3qvx)**

We agree that identifying the most plausible hypothesis is important. However, enumerating all admissible explanations remains essential, because different explanations can be equally consistent with the observations at a given moment. For example, one problem might have multiple equally plausible explanations; as Spirtes et al. [R1] note, “If two causal structures can equally account for the same statistics, then no statistics can distinguish them.”

[R1] Spirtes, P. et. al, Causation, Prediction, and Search, 2000.

---

**Why fully enumerable tasks (->Reviewer oycp, 3qvx, Et1B)**
The intention of fully enumerable tasks is to obtain precise, unambiguous diagnostics of hypothesis-set generation. If the ground-truth space is not enumerable, benchmarking becomes unreliable because coverage of the generated hypotheses cannot be objectively assessed.

---

**Connection to real scientific problems  (->Reviewer oycp, Et1B, 1mJR)**
Each of task abstracts a real-world scientific scenario where multiple explanations coexist:

- **3D understanding** In real-world robot navigation, limited observations can correspond to multiple plausible spatial layouts, a well-known challenge in SLAM [R2]. Our 3D understanding task simulates this geometric underdetermination to evaluate whether existing LLMs can generate and enumerate all admissible scene layouts from limited views.
- **Causal reasoning**  In cell-signaling [R3], edge directions are typically identified through targeted interventions. However, when interventions are limited, multiple causal structures remain equally compatible with the observed correlations. Our causal task abstracts this underdetermination and quantifies it using VR/NR/RR.
- **Genetic interactions** In biological discovery [R4], limited phenotypic observations often admit multiple mechanistically distinct genetic‐interaction mechanisms; our Boolean task models this ambiguity directly.

To further demonstrate, we conducted a complex real-world genetic interaction experiment. We used vesicle trafficking module from a real-world dataset [R4], which includes 6 genes and 14 knockout experiments (6 single, 8 double). To avoid data leakage, we anonymized the gene name as A-F. Given this anonymized dataset, the LLMs were tasked to propose hypotheses for cellular viability.

As indicated in Table R1, performance on real-world data is comparable to synthetic benchmarks. Four SOTA models achieved perfect performance with 100% VR and 70% NR, generating 7 unique valid hypotheses that converge to a consistent solution space, while three failed models demonstrated high diversity (80-100% NR) but 0% validity, indicating they explored extensively but in incorrect directions inconsistent with real biological observations.

| Model | VR | NR | No. of Unique Valid Hypotheses |
|-------|-----|--------|------------------------|
| DeepSeek-R1 | 100% | 70% | 7 |
| Gemini-2.5-Pro | 100% | 70% | 7 |
| GPT-5 | 100% | 70% | 7 |
| Grok-4 | 100% | 70% | 7 |
| Claude Opus-4 | 0% | 100% | 0 |
| GPT-4o | 0% | 100% | 0 |
| Llama-3.3-70b | 0% | 80% | 0 |

Table R1. Real-world genetic interaction experiment results.

[R2] Durrant-Whyte, H., et al. Simultaneous localization and mapping: part I. IEEE robotics & automation magazine, 2006.

[R3] Sachs, Karen, et al. Causal protein-signaling networks derived from multiparameter single-cell data, Science, 2005.

[R4] Costanzo M, et al. A global genetic interaction network maps a wiring diagram of cellular function, Science, 2016.

---

**Connection with Domain Knowledge (-> Reviewer oycp, 3qvx, Et1B)**

Validity Rate (VR) does not quantify diversity. It only measures whether generated hypotheses are consistent with observations. High VR simply indicates that models possess the necessary domain knowledge; if all answers are correct but identical, VR would still be 100%.

In contrast, our Uniqueness/Novelty Rate (NR) evaluates whether hypotheses are mutually non-redundant, and our Recovery Rate (RR) measures coverage over the admissible hypothesis space. These metrics directly capture exploration and creativity rather than correctness alone.

---

### Meta-Review · Area_Chair_oXBw · 2026-01-06

**Summary:**

The submission proposes HypoSpace, a benchmarking framework for evaluating large language models as set-valued hypothesis generators under underdetermined inference. Reviewers raised concerns primarily about:

(i) Motivation and problem formulation, questioning whether generating multiple hypotheses is intrinsically valuable compared to identifying a single plausible explanation, and how VR/NR/RR translate to practical reasoning competence (Reviewers oycp);

(ii) Task realism and external validity, noting that the evaluation domains are highly simplified and may not faithfully reflect real-world scientific reasoning or discovery tasks (Reviewers oycp, Et1B, 1mJR);

(iii) Novelty of empirical findings, arguing that mode collapse of solutions in larger solution spaces has been previously reported, and the proposed method only focused on the extreme simplifications of the complex problems, which is misleading (Reviewers 1mJR);

(iv) Discriminatory power and difficulty scaling, with concerns that some tasks saturate for strong models, potentially limiting their ability to distinguish reasoning capabilities (Reviewers 1mJR);

While the rebuttal addresses several technical and interpretative points, disagreement remains regarding the extent to which results on fully enumerable, abstracted tasks can support broader claims about LLM reasoning and creativity in real-world settings.

**Reviewer Concerns:**

Concerns addressed by the rebuttal include discriminatory power and difficulty scaling, while the following concerns remain outstanding.
- Motivation and practical relevance of multi-hypothesis generation:
 Despite the additional explanations, concerns remain regarding whether generating and enumerating large sets of hypotheses is intrinsically valuable in most real-world reasoning scenarios, and whether VR/NR/RR provide a convincing proxy for practical reasoning competence beyond carefully constructed underdetermined settings.
- Task realism and external validity:
 Although the authors provide analogies to real scientific problems and add a small real-world genetic interaction case study, the evaluation remains centered on highly simplified, fully enumerable domains. It remains unclear to what extent performance on these abstracted tasks meaningfully reflects progress on real-world scientific reasoning or discovery problems.
- Novelty of empirical findings:
 The rebuttal distinguishes the proposed diagnostics from prior work. However, the concern that the main empirical observation has been reported in prior literature is not fully resolved, and the contribution may be viewed as a more precise measurement of an existing phenomenon rather than a fundamentally new insight.

**Reviewer Scores:**

**Reviewer oycp (Score: 2, reject)**

 This reviewer did not provide a follow-up. While the rebuttal addresses several conceptual and empirical points, the core concerns are only partially resolved. A slight positive adjustment is possible, but the assessment would likely remain in the borderline range.

**Reviewer 3qvx (Score: 6, marginally above acceptance)**

 This reviewer indicated willingness to raise the score if questions were addressed. The rebuttal directly responds to questions and a slight positive adjustment is possible.

**Reviewer Et1B (Score: 4, marginally below acceptance)**

 This reviewer explicitly stated that their score would be maintained after the rebuttal. While some concerns were addressed through additional discussion and experiments, the final assessment remains unchanged.

**Reviewer 1mJR (Score: 2, reject)**

 This reviewer provided a follow-up stating that, despite additional experiments on real-world biological data, concerns about the disconnect between the benchmark and real-world scientific progress remain. The score was explicitly maintained.

---

### Decision · Program_Chairs · 2026-01-26

Reject